# Performance evaluation of developed macrophyte-assisted vermifiltration system designed with varied macrophytes and earthworm species for domestic wastewater treatment

**Rapheal Nsiah-Gyambibi**[1,2]*, **Emmanuel Acheampong**[2], **Elizabeth Von-Kiti**[2], **Christian Larbi Ayisi**[3]

1 Department of Civil Engineering, Regional Water and Environmental Sanitation Centre, UPO, College of Engineering Kwame Nkrumah University of Science and Technology, Kumasi, Ghana, 2 Institute of Industrial Research, Council for Scientific and Industrial Research, Accra, Ghana, 3 Department of Water Resource and Sustainable Development, School of Sustainable Development, PMB, University of Environment and Sustainable Development, Somanya, Ghana

* rngyambibi@csir-iir.com, nsiahrapheal@yahoo.com

**Data Availability Statement:** All relevant data are within the paper and its Supporting information files.

## Abstract

Development of sustainable technology to treat domestic wastewater with added advantages of cost reduction and improved handling efficiency is crucial in developing countries. This is because, domestic wastewater from households are stored in septic tanks and are poorly treated prior discharge. This study developed a macrophyte-assisted vermifiltration (MAV) system to treat domestic wastewater. The MAV system is an integrated approach of macrophytes and earthworms in a vermifiltration and complex physicochemical mechanism processes. The use of different macrophyte and earthworm species was hypothesized by the study to affect and vary the treatment performance of the developed MAV. The study therefore aimed to evaluate the treatment performance of the developed MAV when three varied macrophyte species (*Eichhornia crassopes*, *Pistia stratiotes* and *Spirodela sp.*) and two varied earthworm species (*Eisenia fetida* and *Eudrilus eugeniae*) were used to design the treatment system. Treated effluents were collected every 48hours within two weeks for physico-chemical, pathogen and helminth analysis. The contaminants ($N_{tot}$, $NH_3$, $NO_3$-N and $P_{tot}$) in the wastewater were high (>50 $mgL^{-1}$, >5 $mgL^{-1}$, >1 $mgL^{-1}$ and >20 $mgL^{-1}$ respectively). Results revealed that the developed MAV systems were effective in the removal of solids (>60%), nutrients (>60%) and pathogens (>90%). In most cases, there were no significant differences between the selected varied macrophytes and earthworms in the treatment performances. Results therefore demonstrated that the selected macrophytes combined with the earthworm species were suitable when used in the development of the MAV system. Developing the MAV with the selected varied macrophyte and earthworm species did not only contribute to the treatment of the wastewater, but also improved the vermiculture. *Eudrilus eugeniae* however demonstrated higher biomass gain (5–10% more) compared to *Eisenia fetida*.

**Funding:** The authors received no specific funding for this work.

**Competing interests:** The authors have declared that no competing interests exist.

## 1. Introduction

In the tropical regions like Ghana, most of the domestic wastewater from households are collected and stored in septic tanks [1]. These wastewater are poorly treated and infiltrate the subsurface to contaminate the groundwater table, contributing to freshwater depletion [2]. Contaminants in the wastewater such as organic and inorganic solids, nutrients as well as pathogens, have been reported to cause serious health effects [3]. The contaminants remain either in soluble, colloidal or suspended form. The need to remove these contaminants prior disposal is very crucial [4]. Contaminants such as nitrate, sulfate, phosphate, fluoride, chloride and oxalate have some hazardous health effects [5]. For example, high concentration of nitrate in children causes blue babies disease (methemoglobinemia) while it is well-known that high concentration of fluoride results to fluorosis [6].

The general aim of domestic wastewater treatment focuses on safe discharge and reuse. However, domestic wastewater treatment involving conventional approaches seems unrealistic for many developing countries [1]. This has made it necessary to consider low-cost options which are low-energy demanding and able to meet treatment objectives [7]. That is, there is an urgent need to develop sustainable treatment strategies with added advantages of cost reduction and handling efficiency. Filtration and sedimentation are among the economical and effective wastewater treatment techniques that primarily depends on the combination of complex physicochemical mechanisms [8]. In terms of biological filtration, vermifiltration is gaining wide popularity [9–12]. Vermifiltration, which integrates earthworm and microbes, is one of the filtration techniques gaining much attention in wastewater treatment due to its low energy consumption, less operation, less maintenance costs and chemical-free treatment process [13]. The earthworm species commonly used in vermifiltration processes are *Eisenia fetida* and *Eudrilus eugeniae* [14–16]. The resulting treated wastewater are potential sources for reuse such as irrigation [17]. Vermifiltration technology has evolved into the incorporation of macrophytes. This technique is known as macrophyte-assisted vermifiltration (MAV). The introduction of macrophytes into MAV systems seeks to improve the contaminant removal and address the problem of resource recovery by converting nutrients present in wastewater into reusable forms [18]. MAV is an innovative, holistic, and sustainable approach that has been the subject of recent initiatives. Reliance on macrophytes to treat wastewater in vermifiltration is gaining recognition worldwide because of their ability to remove nutrients and their potential use for agricultural and other economic activities [17]. MAV provides a cyclic treatment and an integrated approach to wastewater management. Nutrients are taken up by macrophytes which, when harvested, can be used to feed livestock or fish, and the effluents can be used for agricultural purposes such as irrigation [19]. There are several species of aquatic plants that can be used as macrophytes in MAV systems [19].

However, factors influencing macrophyte selection in MAV have not been adequately investigated. This has left a research gap in identifying local conditions that influences the use of varied macrophytes in MAVs. For example, in Ghana, Water Hyacinth (*Eichhornia crassopes*), Water Lettuce (*Pistia stratiotes*) and Duckweed (*Spirodela sp.*) are readily available and pose no serious threat to the local aquatic environment, but their efficiencies in contaminant removal in MAV is inadequately reported [1]. These macrophytes have several intrinsic properties that makes them indispensable components for wastewater treatment [20]. Investigations into the use of varied earthworm species combining with different macrophytes to improve contaminant removal in MAV systems is of crucially importance for optimum performances. Research knowledge in this research gap could provide basis for the development of this innovative MAV systems, which could be integrated as secondary treatment units to household septic tanks. There is also a greater scope for the application of this

research knowledge in managing domestic wastewater for reuse. This study therefore investigated the use of developed MAV systems for domestic wastewater treatment using three different macrophytes (Water Hyacinth (*Eichhornia crassopes*), Water Lettuce (*Pistia stratiotes*) and Duckweed (*Spirodela sp.*)) combined with two varied earthworm species (*Eudrilus eugeniae* and *Eisenia foetida*).

## 2. Methods

### 2.1. Study location

The study was carried out at the Environmental Engineering Laboratory located in Kumasi. Kumasi is a city in the Ashanti region of Ghana with a tropical forest belt between latitude 6.400 and 6.350 N and longitude 1.30 and 1.35 W. Kumasi is at 250-399m above sea level with an average ambient temperature of 25–28˚C which is the optimum temperature range for earthworm species [21]. The experiments were performed during the months of April–May.

### 2.2. Experimental setup

Fresh domestic wastewater was obtained from a household septic tank within the Oforikrom sub-metro in Kumasi Metropolitan Assembly (KMA) located in Ghana. Three different macrophytes and two different earthworm species (based on their availability) were used in the study. The macrophytes consisted of Water Hyacinth (*Eichhornia crassopes*), Water Lettuce (*Pistia stratiotes*) and Duckweed (*Spirodela sp.*) while the earthworm species consisted of *Eudrilus eugeniae* and *Eisenia foetida*. Six different MAV experimental treatment setups were then constructed. The setups were labelled HE, LE, DE, HF, LF and DF. Thus, HE had Water Hyacinth and *Eudrilus eugeniae*, LE had Water Lettuce and *Eudrilus eugeniae*, DE had Duckweed and *Eudrilus eugeniae*, HF had Water Hyacinth and *Eisenia foetida*, LF had Water Lettuce and *Eisenia foetida* and DF had Duckweed and *Eisenia foetida*. A setup consisting of an empty barrel, which only contained the wastewater to mimic the conditions in septic tanks, was labelled as the control. The macrophytes used were obtained from a stream and cultured in tap water for 24 hours following the procedure used in [22]. About 0.25m$^2$ patches of cultured macrophytes were stocked in the macrophyte chamber of the experimental treatment units following the stocking density described in [23]. The earthworm species used for the vermiculture were obtained from a breeding stock cultured in the laboratory at a temperature of 25˚C. The macrophyte chamber and the vermifilter bed of each setup were constructed with 60.0 cm$^3$ polyethylene terephthalate barrels following design specifications described in [17]. Schematic diagram of the setup is presented in Fig 1. The vermifilter bed had three layers. The top layer (50 cm thick) was made up of coconut coir (6–8 mm) as a bulking material with an empty space of 5 cm at the top for aeration purpose. The middle layer (55 cm thick) consisted of sand (1–2 mm, 10 cm thick), gravels (6–8 mm) and matured vermicompost as a substrate. This middle layer housed the earthworm packing bed where four hundred and fifty (live weight ~255–275 mg) clitellated earthworms species were added, following the stocking density used by [24]. The bottom was the supporting layer and consisted of coarse layer of lateritic hardpan gravels (12–14 mm, 15 cm thick). The experimental setups were allowed to acclimatize for seven days before the start of the experiments. The wastewater was pumped into the macrophyte chamber using a 0.5HP single stage laboratory vacuum pump at a hydraulic loading rate (HLR) of 0.339 m$^3$ m$^{-2}$ d$^{-1}$. HRL is critical for the optimal treatment performance of MAVs and this HRL used was suitable to prevent clogging in the treatment setups. Infiltration of effluents from the macrophyte chamber through the vermibed occurs by gravitational flow in a vertical flow system (VFS) through a showerhead of 1–2mm perforations for its uniform distribution. Following the sampling procedures in [1,17], effluent samples after macrophyte

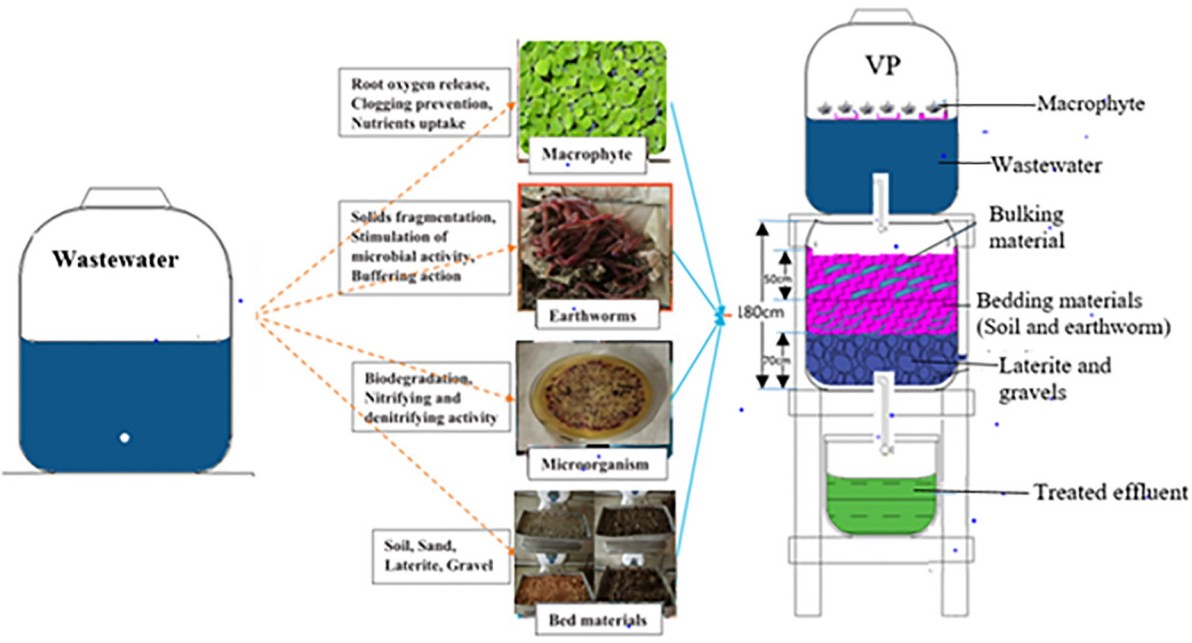

**Fig 1.**

chamber and vermifiltration were collected every 48hours for two weeks for physico-chemical, pathogen and helminth analysis.

## 2.3. Physico-chemical and heavy metal analysis

The physico-chemical parameters assessed were temperature, pH, electrical conductivity (EC), total dissolve solids (TDS), total suspended solids (TSS), turbidity, dissolve oxygen (DO), chemical oxygen demand (COD), biochemical oxygen demand (BOD), total nitrogen ($N_{tot}$), ammonia ($NH_3$), ionized ammonia ($NH_4$-N), nitrate ($NO_3$-N), total phosphorus ($P_{tot}$), chlorides and dissolved organic carbon (DOC). pH, EC, turbidity, TDS and DO were measured using Palintest multi portable meter. TSS of samples were determined following the standard protocol of filtration and gravimetric oven drying method [25]. COD assessments were made using high-range ampoules (HACH Chemical) with a spectrophotometer (HACH, DR5000) [26]. BOD was assessed following the Lovibond Water Testing protocol of BOD-System BD 600 method. $N_{tot}$ was assessed using Persulfate Digestion Method 10072 of the TNT protocol. $P_{tot}$ was assessed using the Molybdovanadate with Acid Persulfate Digestion Hach Method 10127 (Hach Company, Loveland, Colorado, USA). $NH_3$, $NH_4$-N, $NO_3$-N, chlorides and DOC were assessed following Standard Methods [25].

The contaminant percentage reduction was calculated using the formula:

$$Percentage\ reduction\ (\%R) = \frac{(C_i - C_o)}{C_i} * 100$$

where $C_i$ and $C_o$ represent the influent value and effluent value, respectively.

## 2.4. Pathogen analysis

The indicator organisms for the pathogen assessment was total coliform, faecal coliform and *E. coli*. The indicator organisms for the helminths assessment was Ascaries, Tapeworms and

*Fasciola* sp. The pathogen assessment was conducted following the colony forming unit (CFU) standard plate count technique using Membrane filtration and Chromocult agar isolation method [25]. The helminth assessment was conducted using the Microscopic examination method [25].

The pathogen and helminth removal efficiency was calculated using the formula:

$$Percentage\ removal\ (\%R) = \frac{(C_{in} - C_{out})}{C_{in}} * 100 \tag{1}$$

where $C_{in}$ and $C_{out}$ represent the influent pathogen or helminth concentration and effluent pathogen or helminth concentration, respectively.

## 2.5 Vermiculture

Two earthworm species (*Eudrilus eugeniae* and *Eisenia foetida*) were used. The initial individual live weight of earthworms was determined with electronic scale prior stocking into vermibeds. This was done to monitor the growth of the earthworms. The earthworms' growth and cocoon production in each treatment unit were observed at the end of the study. Newly hatched cocoons (<255 mg) were counted as cocoon production. The earthworms and the cocoons were separated from the composted material by hand sorting and washed in tap water to remove adhering material before weighing. The washed earthworms were weighed on a live weight basis in a water filled weighing basin to prevent the worms from desiccating which could have affected their weight. All measured earthworms were returned to their respective containers and the cocoons were counted and introduced into separate bedding. Earthworm biomass in the form of growth rate (mg day$^{-1}$) and individual reproduction rate (cocoonworm$^{-1}$ day$^{-1}$) were estimated using Eqs 2 and 3:

*Growth rate (GW)* was estimated using Eq 1

$$GW = \left(\frac{\text{Earthworm growth}}{\text{Number of days}}\right) \tag{2}$$

*Individual reproduction rate (IRP)* was estimated using Eq 3

$$IRP = \left(\frac{\text{Number of cooncoons}}{\text{toatal number of earthworms} \times \text{Number of days}}\right) \tag{3}$$

## 2.6. Statistical analysis

Chi-square test of homogeneity and Shapiro-Wilk normality test were conducted to determine the homogeneous and parametric nature of the variables before performing ANOVA. ANOVA tests were conducted to determine the interaction effect between variables and when significant effects were detected, the groups were subjected to post-hoc Tukey's HSD test. The level of significance used for all statistical tests was 5% ($p < 0.05$).

## 3. Results

### 3.1. Physico-chemical changes

The average temperature of the wastewater was above the room temperature of 25°C (Table 1). Effluents from the macrophyte chamber and vermifiltration of experimental setups recorded drops in this temperature (Fig 2A). However, these changes showed no significant variations ($p > 0.05$) when compared to the control. The pH of the wastewater ranged from

**Table 1. Physicochemical, pathogen and helminth characteristics of wastewater (mean ± S.D.; n = 3).**

| Parameter | Value |
|---|---|
| **Physico-chemical** | |
| Temperature (˚C) | 27.33 ±0.29 |
| pH | 8.1–8.3 |
| Electrical conductivity, EC ($\mu scm^{-1}$) | 3233.33 ±251.66 |
| Total dissolved solids, TDS ($mgL^{-1}$) | 1700.67 ±263.44 |
| Total suspended solids, TSS ($mgL^{-1}$) | 255.67 ±41.67 |
| Total Nitrogen, $N_{tot}$ ($mgL^{-1}$) | 78.00 ±6.08 |
| Ammonia, $NH_3$ ($mgL^{-1}$) | 52.00 ±2.65 |
| Ionized ammonia, $NH_4$-N ($mgL^{-1}$) | 9.67 ±2.08 |
| Nitrate, $NO_3$-N ($mgL^{-1}$) | 2.07 ±0.25 |
| Total Phosphorus, $P_{tot}$ ($mgL^{-1}$) | 37.40 ±4.13 |
| Chlorides, ($mgL^{-1}$) | 48.00 ±4.00 |
| Dissolved oxygen, DO ($mgL^{-1}$) | 0.03 ±0.06 |
| Biochemical oxygen demand, BOD ($mgL^{-1}$) | 120.00 ±10.00 |
| Chemical oxygen demand, COD ($mgL^{-1}$) | 240.00 ±26.46 |
| Turbidity (NTU) | 265.67 ±8.14 |
| Dissolved organic carbon, DOC ($mgL^{-1}$) | 104.00 ±6.56 |
| **Pathogen** | |
| Faecal bacteria (CFU 100 ml[-1]) | 1.26E+07 ±2.05E+07 |
| Total coliform (CFU 100 ml[-1]) | 1.80E+08 ±2.95E+08 |
| *Escherichia coli* (CFU 100 ml[-1]) | 1.28E+07 ±1.11E+07 |
| **Helminth eggs** | |
| Ascaries (CFU 100 ml[-1]) | 2.33E+03 ±3.84E+03 |
| Tapeworms (CFU 100 ml[-1]) | 4.85E+03 ±7.92E+03 |
| *Fasciola sp* (CFU 100 ml[-1]) | 8.51E+02 ±1.31E+03 |

8.1 to 8.3 and results revealed that the pH steadily reduced towards the neutral in all treated effluents (Fig 2B). pH reductions in the treated effluents showed significant variations ($p<0.05$) at all sampling points except the 48hour sampling period. The lowest percentage pH reduction of 5.82 ±0.3% was recorded in the control while HE demonstrated the highest percentage pH reduction (9.16 ±0.6% and 11.88 ±0.4% in the macrophyte chamber effluents and vermifiltrate respectfully). The pH values in the effluents obtained from the bio-filters at the end of the study were within WHO effluent discharge standard of 6.5–8.5 [27].

The electrical conductivity (EC) of the wastewater was high (>2500 $\mu scm^{-1}$) and this steadily reduced in all treated effluents (Fig 2C). The control demonstrated the least percentage EC reductions of 75.77 ±0.2% compared to effluents from the macrophyte chamber and vermifiltrate of treatment setups. Thus, HE (88.70 ±0.2% and 96.59 ±0.4% respectively), LE (88.59 ±0.2% and 96.27 ±0.4% respectively), DE (88.94 ±0.3% and 96.30 ±0.2% respectively), HF (88.75 ±0.2% and 96.27 ±0.2% respectively), LF (88.85% ±0.4 and 96.29 ±0.2% respectively) and DF (88.83 ±0.3% and 96.32 ±0.2% respectively). These reductions in the treatments setups were significant ($p<0.05$) when compared to the control at all sampling points.

TDS in the wastewater were equally high (>1500 $mgl^{-1}$) and this steadily reduced at the end of the study (Fig 3A). The control recorded the least percentage reduction of 56.21 ±0.5% compared to the effluents of the treatment setups (78%– 80% in the macrophyte chamber effluents and 93%– 96% in vermifitrates). These reductions in the treatments setups were significant ($p<0.05$) when compared to the reduction in the control in all sampling

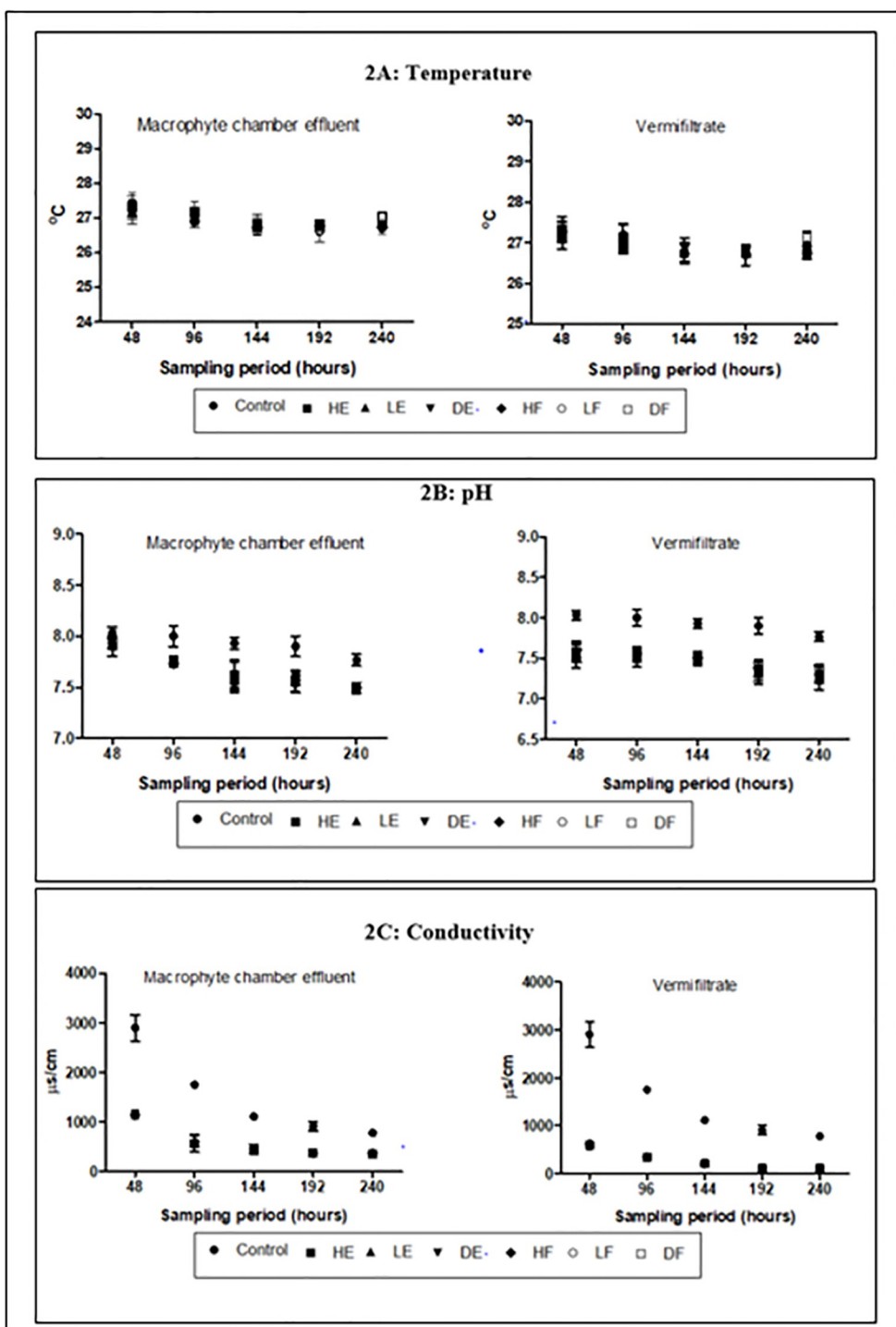

**Fig 2. Physicochemical characteristics of effluent (A) Temperature (B) pH (C) Conductivity.**

points. TSS in the wastewater followed similar trend (Fig 3B). High TSS (>200 mgl⁻¹) was recorded in the wastewater and this significantly reduced ($p < 0.05$) at the end of the study in the effluent of all experimental setups. However, the control again had the least reduction (25.03 ±0.2%) compared to the reductions in the effluents of the treatment setups (60%–

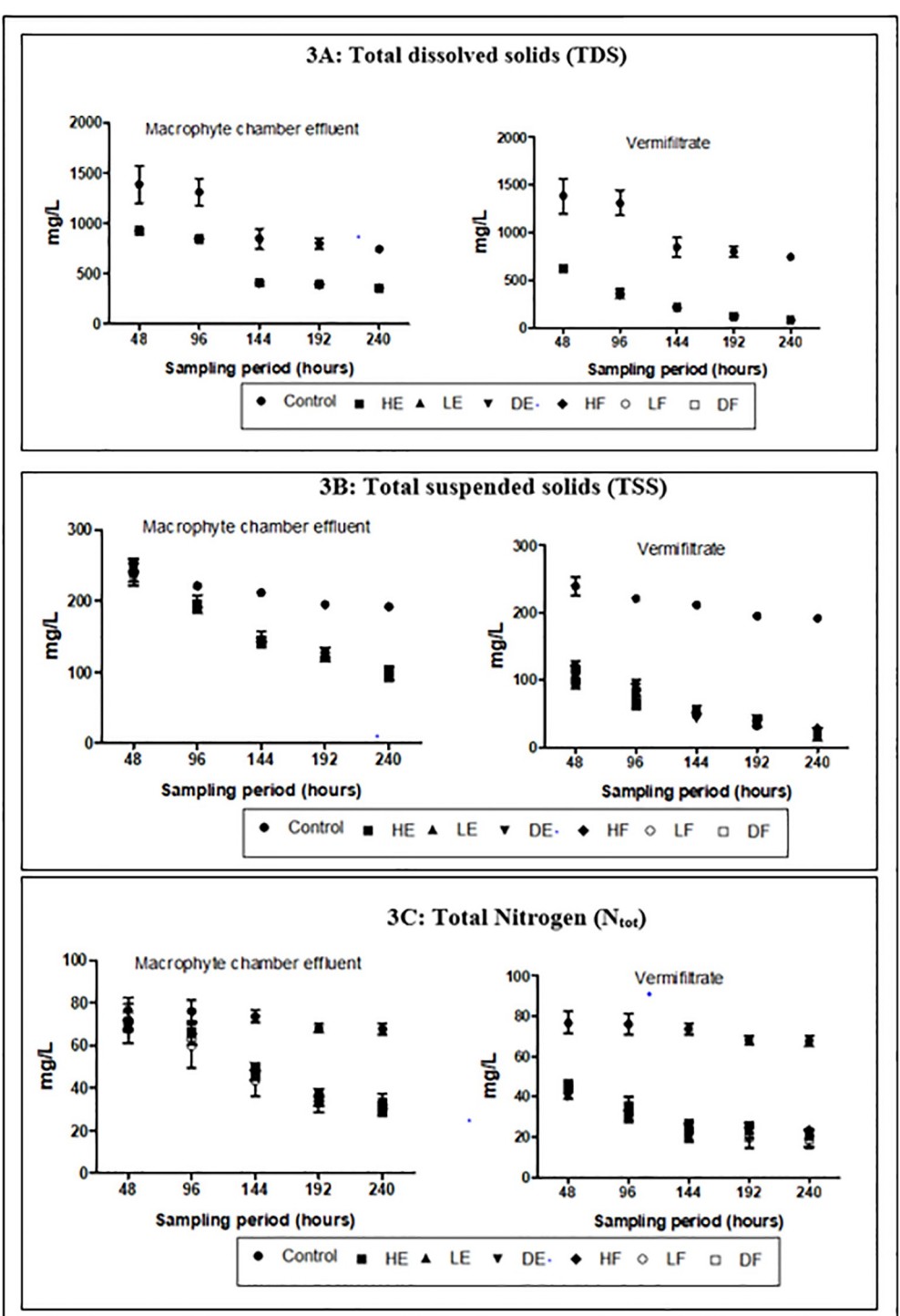

**Fig 3. Physicochemical characteristics of effluent (A) TDS (B) TSS (C) $N_{tot}$.**

62% in the macrophyte chamber effluents and 90%–94% in vermifiltrates). Aside the first sampling points in the macrophyte chamber that showed no significant reductions ($p > 0.05$), the reductions in the experimental setup effluents were significant ($p < 0.05$) in all sampling points when compared to the control.

The mean concentrations for $N_{tot}$, $NH_3$, $NO_3$-N and $P_{tot}$ in the wastewater as presented in Table 1 were high (>50 mgl$^{-1}$, >5 mgl$^{-1}$, >1 mgl$^{-1}$ and >20 mgl$^{-1}$ respectively) while ionized ammonia ($NH_4$-N) was low (<10 mgL$^{-1}$). During the entire experiment, the mean concentrations of these nutrients received varied changes in the effluents (Figs 3C, 4A–4C and 5A). The

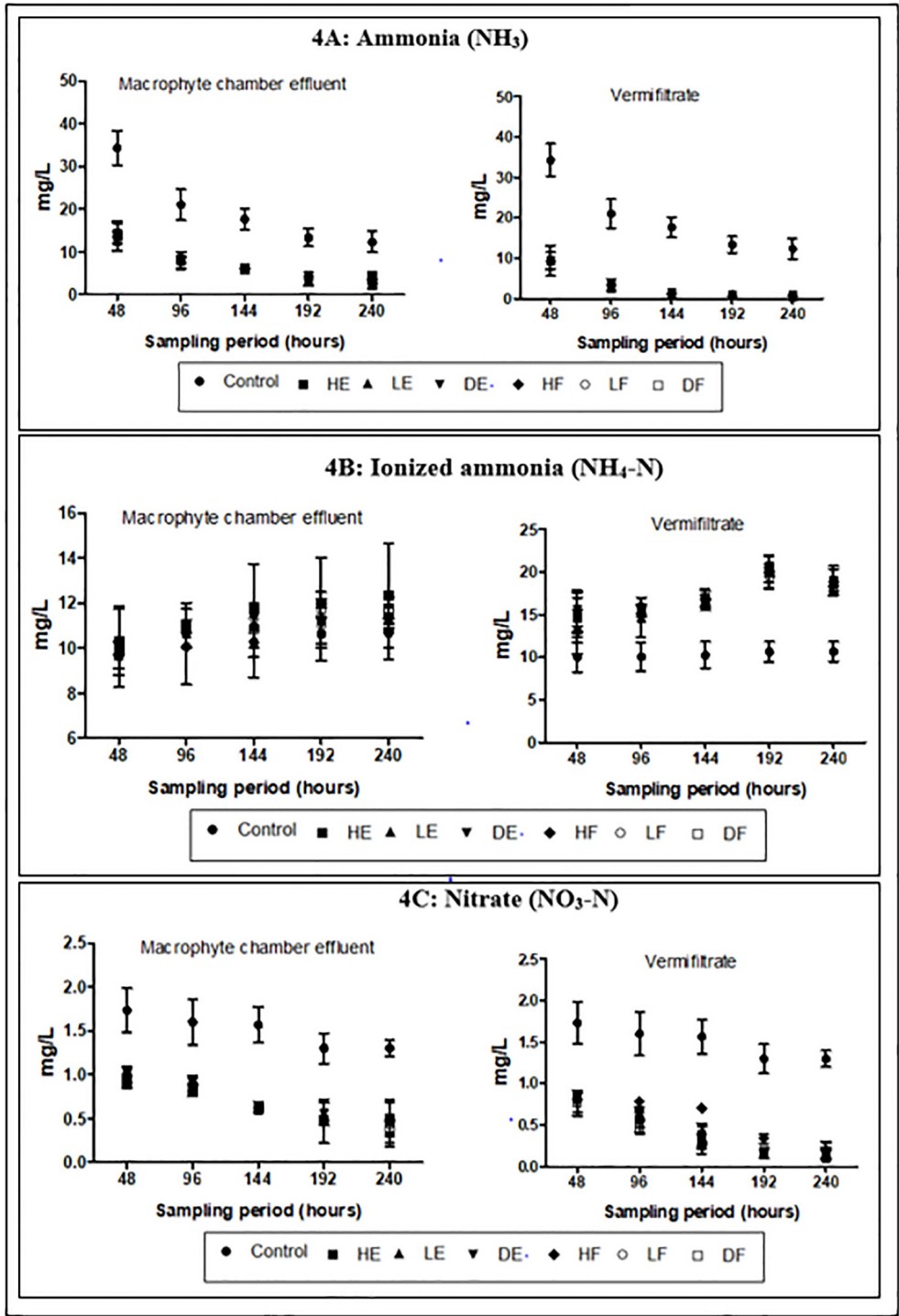

**Fig 4. Physicochemical characteristics of effluent (A) NH$_3$ (B) NH$_4$-N (C) NO$_3$-N.**

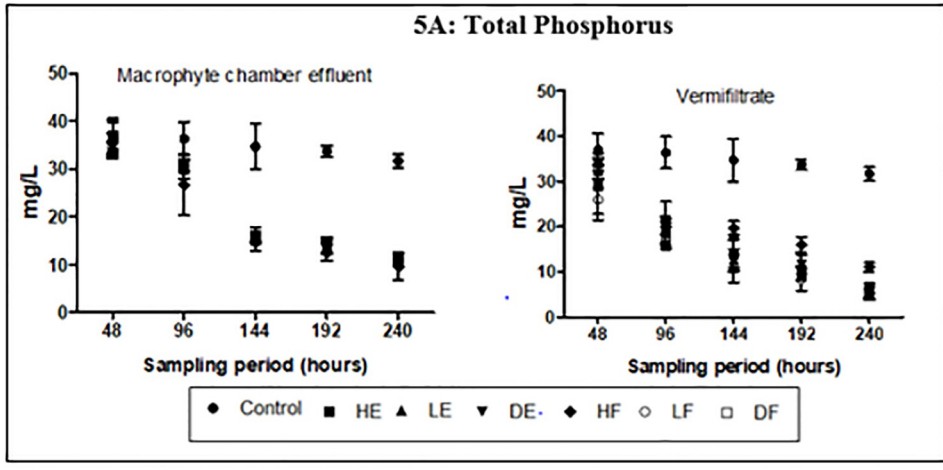

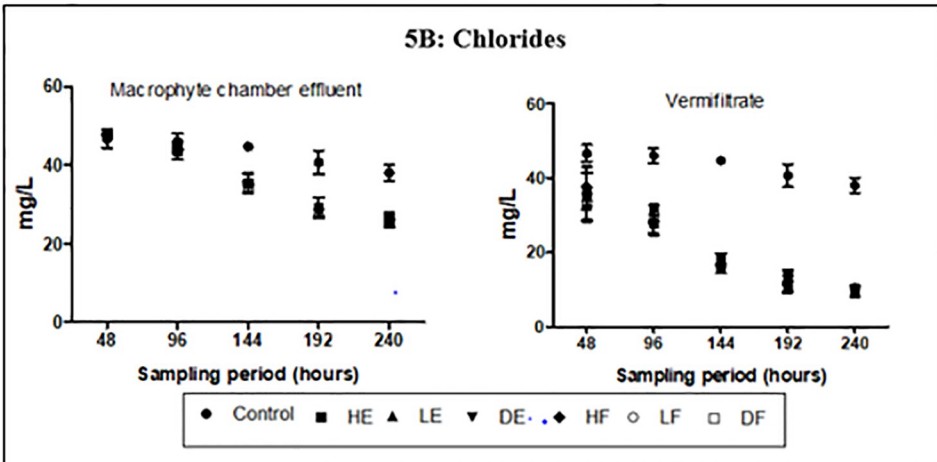

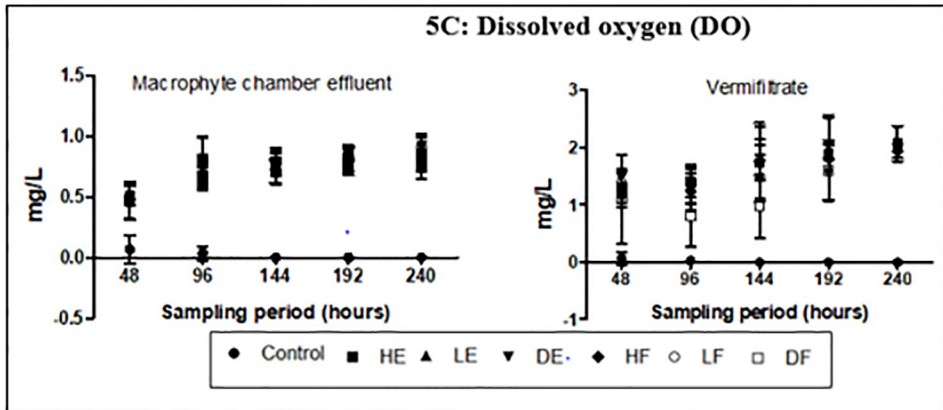

**Fig 5. Physicochemical characteristics of effluent (A) Total phosphorus (B) Chlorides (C) DO.**

control demonstrated 13.25%, 76.28%, 32.37% and 15.33% reductions in $N_{tot}$, $NH_3$, $NO_3$-N and $P_{tot}$ respectively. These percentage reductions in the control were the least recorded when compared to the effluents from the experimental treatments. The percentage reductions of $N_{tot}$ in the macrophyte chamber and vermifiltration effluents were 58%– 60% and 71%– 77% respectively, $NH_3$ were 71%– 77% and 92%– 98% respectively, $NO_3$-N were 77%– 89% and

90–96% respectively and 68%– 72% and $P_{tot}$ were 82–87% respectively. $NH_4$-N however recorded an increase. The control recorded the least increase of 10.34 ±0.4% while HE recorded the highest increase in the macrophyte chamber and vermifiltration (27.59 ±0.2% and 96.55 ±0.3% respectively). The increase followed the following order; LF (20.43 ±0.2% and 95.17 ±0.2% respectively), LE (20.39 ±0.3% and 94.07 ±0.2% respectively), DF (19.92 ±0.3% and 90.80 ±0.2% respectively), HF (19.92 ±0.3% and 89.66 ±0.2% respectively) and DE (18.39 ±0.2% and 89.60 ±0.4% respectively). Chlorides in the wastewater was high ($>20mgL^{-1}$) and these were reduced in the treated effluents (Fig 5B). The control showed the least percentage reduction (20.87 ±0.2%) and the highest in DE (46.99 ±0.5% and 81.25 ±0.3%) followed by DF (47.45 ±0.3% and 81.02 ±0.2%), LE (45.14 ±0.3% and 80.56 ±0.3%), HE (45.83 ±0.2% and 79.86 ±0.3%), HF (45.99 ±0.2% and 78.70 ±0.3%) and LE (27.59 ±0.2% and 96.55 ±0.3%). The wastewater was highly turbid ($>150$ $mgL^{-1}$) and had high DOC ($>50$ $mgL^{-1}$). The treatment process significantly reduced the turbidity ($p<0.05$) and the DOC ($p<0.05$). Again, the control recorded the least percentage reduction in the turbidity (18.44 ±0.3%) and DOC (18.91 ±0.3%) compared to the effluents of the treatment setups (Figs 5C and 6D respectively). With respect to dissolved oxygen (DO), the control recorded the least DO compared to the effluents of the treatment setups (Fig 5C). The treatment setups showed higher significant DO improvement performances ($p<0.05$) than the control at all sampling points. The BOD and the COD were equally high in the wastewater ($>50$ $mgL^{-1}$ and $>100$ $mgL^{-1}$ respectively) but both parameters significantly reduced ($p<0.05$) in the treated effluents at the end of the study.

## 3.2. Pathogen removal

Pathogen and helminth analysis carried out for the wastewater showed high levels of all indicator organism contamination (Table 2). Overall, removal of pathogen indicator was $<50\%$ and $>90\%$ in the control and the treatment effluents respectively. The reduction of the three groups of pathogen (Faecal coliform, total coliform and *E. coli*) generated K values of 1.68– 1.70. The control recorded the least pathogen removal ($>20\%$). The indicators of the helminth (Ascaries, Tapeworms and *Fasciola* sp) recorded similar percentage removal and statistically, there was significant variation ($p < 0.05$) between treatment effluents when compared to the control.

## 3.3. Vermiculture

The growth and reproduction rate of the earthworm species in different treatments are presented in Table 3. Earthworm mortalities were less than 10% in all treatments. *Eudrilus eugeniae* treatments however demonstrated lower mortalities compared to *Eisenia foetida* treatments, but these differences were not significant ($p>0.05$). Meanwhile, *Eudrilus eugeniae* demonstrated significant higher individual weight gain ($p<0.05$) compared to *Eisenia foetida*. Results also showed *Eudrilus eugeniae* demonstrated significant higher total individual live weight ($p<0.05$), individual weight gain ($p<0.05$), biomass gain ($p<0.05$), individual growth rate ($p<0.05$) and cocoon production ($p<0.05$). Results demonstrated that earthworm biomass and cocoon production increased in enriched substrates at the end of the study.

## 4. Discussion

### 4.1. Physico-chemical changes

Changes in the pH and electrical conductivity of effluents demonstrated that the macrophytes and the earthworms in the vermibed exhibited natural inherent buffering capacity. The pH reduction trend is similar to results in [28,29]. In the aquatic environment, ions are used by

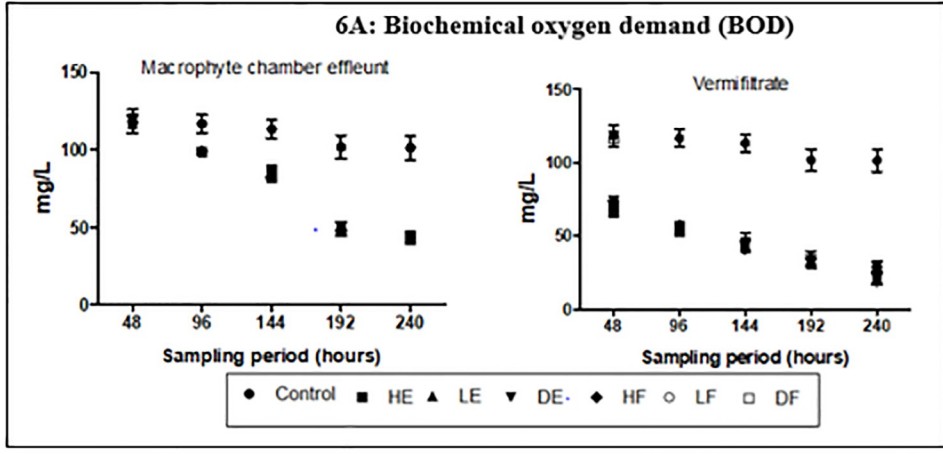

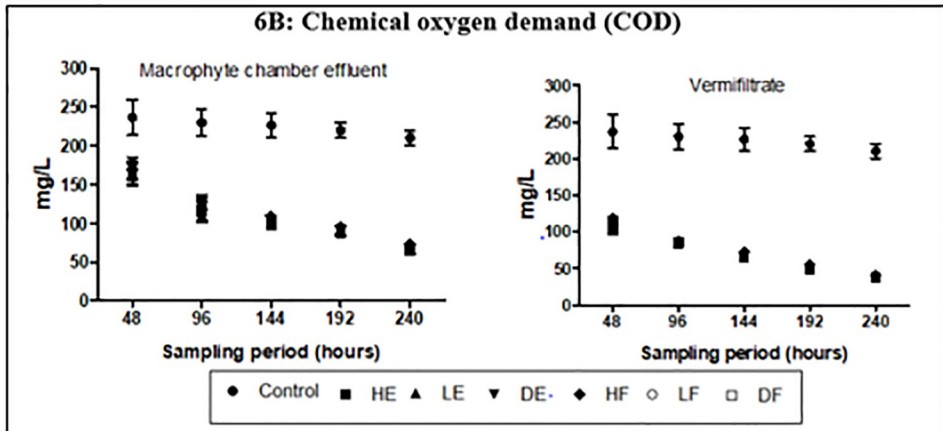

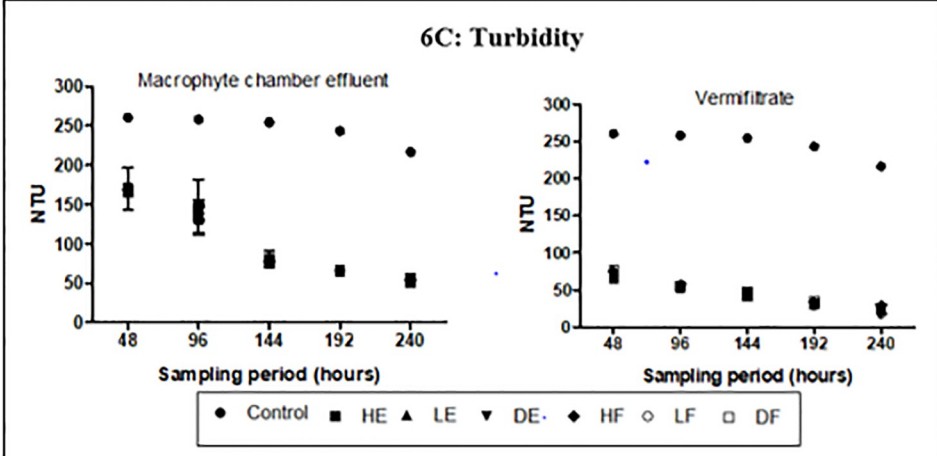

**Fig 6. Physicochemical characteristics of effluent (A) BOD (B) COD (C) Turbidity (D) DOC.**

living tissues in metabolic reactions such as cells development and energy generation [30]. This could explain the significant lowering of pH and EC in the MAV effluents when compared to the control. The physicochemical analysis of the domestic wastewater before treatment revealed very low DO (>1mg/L) and this could be due to the presence of large BOD, COD, solids and nutrient materials. These materials consume and deplete the available DO

**Table 2. Pathogen count in effluents (mean±S.D; n = 3).**

| Treatment | | Faecal coliform (CFU 100 ml$^{-1}$) | Total coliform (CFU 100 ml$^{-1}$) | E. coli (CFU 100 ml$^{-1}$) | Ascaries (CFU 100 ml$^{-1}$) | Tapeworm (CFU 100 ml$^{-1}$) | Fasciola sp (CFU 100 ml$^{-1}$) |
|---|---|---|---|---|---|---|---|
| | Control | $1.53 \times 10^6 \pm 2.46 \times 10^7$ | $2.00 \times 10^7 \pm 2.11 \times 10^8$ | $2.03 \times 10^7 \pm 6.60 \times 10^6$ | $2.08 \times 10^2 \pm 4.70 \times 10^3$ | $5.29 \times 10^2 \pm 6.35 \times 10^3$ | $1.53 \times 10^2 \pm 2.70 \times 10^3$ |
| HE | Macrophyte chamber effluents | $3.63 \times 10^4 \pm 1.53 \times 10^2$ | $5.20 \times 10^4 \pm 2.00 \times 10^2$ | $1.81 \times 10^4 \pm 2.03 \times 10^2$ | $6.77 \times 10^3 \pm 2.08 \times 10^2$ | $1.40 \times 10^4 \pm 5.29 \times 10^2$ | $2.37 \times 10^3 \pm 1.53 \times 10^2$ |
| | Vermifiltrate | $2.63 \times 10^3 \pm 8.89 \times 10^2$ | $2.38 \times 10^3 \pm 8.75 \times 10^2$ | $4.87 \times 10^2 \pm 6.07 \times 10^1$ | $2.75 \times 10^1 \pm 4.41$ | $2.88 \times 10^1 \pm 5.43$ | $3.47 \times 10^1 \pm 2.95$ |
| LE | Macrophyte chamber effluents | $3.43 \times 10^4 \pm 2.08 \times 10^2$ | $5.17 \times 10^4 \pm 1.53 \times 10^2$ | $1.82 \times 10^4 \pm 2.06 \times 10^2$ | $6.83 \times 10^3 \pm 3.21 \times 10^2$ | $1.40 \times 10^4 \pm 5.13 \times 10^2$ | $2.37 \times 10^3 \pm 1.54 \times 10^2$ |
| | Vermifiltrate | $3.33 \times 10^3 \pm 1.55 \times 10^3$ | $3.27 \times 10^3 \pm 9.07 \times 10^2$ | $5.43 \times 10^2 \pm 2.47 \times 10^2$ | $2.27 \times 10^1 \pm 1.03 \times 10^1$ | $3.50 \times 10^1 \pm 1.90 \times 10^1$ | $3.50 \times 10^1 \pm 1.35 \times 10^1$ |
| DE | Macrophyte chamber effluents | $3.53 \times 10^4 \pm 1.00 \times 10^2$ | $5.18 \times 10^4 \pm 1.67 \times 10^2$ | $1.82 \times 10^4 \pm 8.33 \times 10^2$ | $6.80 \times 10^3 \pm 3.33 \times 10^1$ | $1.40 \times 10^4 \pm 1.67 \times 10^1$ | $2.37 \times 10^3 \pm 1.67$ |
| | Vermifiltrate | $1.63 \times 10^3 \pm 9.81 \times 10^2$ | $2.37 \times 10^3 \pm 2.10 \times 10^3$ | $4.94 \times 10^2 \pm 2.26 \times 10^2$ | $3.13 \times 10^1 \pm 1.03 \times 10^1$ | $2.50 \times 10^1 \pm 1.35 \times 10^1$ | $3.17 \times 10^1 \pm 1.80 \times 10^1$ |
| HF | Macrophyte chamber effluents | $3.53 \times 10^4 \pm 1.00 \times 10^2$ | $5.10 \times 10^4 \pm 1.62 \times 10^2$ | $1.62 \times 10^4 \pm 8.30 \times 10^2$ | $6.80 \times 10^3 \pm 3.33 \times 10^1$ | $1.40 \times 10^4 \pm 1.67 \times 10^1$ | $2.37 \times 10^3 \pm 1.67$ |
| | Vermifiltrate | $2.93 \times 10^3 \pm 1.36 \times 10^3$ | $1.52 \times 10^3 \pm 2.15 \times 10^3$ | $4.23 \times 10^2 \pm 1.41 \times 10^2$ | $2.84 \times 10^1 \pm 7.43$ | $2.63 \times 10^1 \pm 1.31 \times 10^1$ | $3.76 \times 10^1 \pm 1.08 \times 10^1$ |
| LF | Macrophyte chamber effluents | $3.47 \times 10^4 \pm 5.77 \times 10^2$ | $5.17 \times 10^4 \pm 9.62 \times 10^2$ | $1.80 \times 10^4 \pm 4.81 \times 10^2$ | $6.82 \times 10^3 \pm 1.92 \times 10^1$ | $1.40 \times 10^4 \pm 9.62$ | $2.37 \times 10^3 \pm 9.62 \times 10^1$ |
| | Vermifiltrate | $1.63 \times 10^3 \pm 9.81 \times 10^2$ | $2.38 \times 10^3 \pm 2.08 \times 10^3$ | $5.67 \times 10^2 \pm 2.06 \times 10^2$ | $3.49 \times 10^1 \pm 1.47 \times 10^1$ | $3.75 \times 10^1 \pm 5.27$ | $4.56 \times 10^1 \pm 7.61$ |
| DF | Macrophyte chamber effluents | $3.57 \times 10^4 \pm 8.82 \times 10^2$ | $5.12 \times 10^4 \pm 1.08 \times 10^2$ | $1.58 \times 10^4 \pm 4.12 \times 10^2$ | $6.76 \times 10^3 \pm 6.12 \times 10^1$ | $1.41 \times 10^4 \pm 2.39 \times 10^2$ | $2.41 \times 10^3 \pm 7.62 \times 10^1$ |
| | Vermifiltrate | $1.63 \times 10^3 \pm 9.81 \times 10^2$ | $2.32 \times 10^3 \pm 2.05 \times 10^3$ | $5.67 \times 10^2 \pm 2.06 \times 10^2$ | $3.49 \times 10^1 \pm 1.47 \times 10^1$ | $3.75 \times 10^1 \pm 5.27$ | $4.56 \times 10^1 \pm 7.61$ |

[31]. The DO in the treated effluents however increased after the macrophyte-vermifiltration and this development could be due to aeration by the actions of the macrophytes and earthworms. The rhizosphere connected to macrophyte roots with active oxygenic photosynthetic characteristics allows the transfer of oxygen to the vicinity of the roots [32]. An important function of macrophytes in the transport of oxygen through the root cells could have facilitated

**Table 3. Earthworm growth and reproduction rate in different treatments (mean ± S.D., n = 3).**

| Treatment | Total earthworm mortality at the end of experiment (%) | Initial individual live weight (mg) | Individual weight gain (%) | Total individual live weight (mg) | Biomass gain (mg) | Individual growth rate (mg day$^{-1}$) | Individual Reproduction rate (cocoonsworm$^{-1}$ day$^{-1}$) |
|---|---|---|---|---|---|---|---|
| HE | 8.33 ±5.77a | 281.42 ±0.71a | 18.13 ±3.21a | 340.40 ±4.50a | 46.24 ±2.20a | 4.19 ±0.15a | 0.067 ±0.011a |
| LE | 8.66 ±2.08a | 281.75 ±0.47a | 17.85 ±5.00a | 333.40 ±1.42b | 45.88 ±4.10b | 4.13 ±0.25a | 0.065 ±0.008a |
| DE | 8.15 ±5.03a | 281.55 ±0.57a | 18.08 ±2.30a | 337.64 ±11.77ab | 46.09 ±2.01ab | 4.18 ±0.21a | 0.064 ±0.005ab |
| HF | 8.30 ±3.40a | 268.89 ±0.35b | 16.60 ±2.12b | 318.80 ±2.45c | 45.86 ±3.20c | 4.04 ±0.36b | 0.062 ±0.01b |
| LF | 8.57 ±3.30a | 270.28 ±0.55b | 16.30 ±3.20b | 313.60 ±4.52c | 46.21 ±2.31c | 4.09 ±0.20b | 0.065 ±0.010a |
| DF | 8.16 ±3.70a | 270.40 ±0.50b | 16.40 ±2.50b | 314.20 ±3.60c | 46.50 ±3.60c | 3.94 ±0.45c | 0.066 ±0.011a |

the decomposition of organic matter and the conversion of ammonium to nitrate [33]. This biochemical process increases the DO concentration. The decomposition of organic matter and conversion of ammonium to nitrate in this same biochemical process could have also attributed to the decrease in DOC and the increase in $NO_3$. Aeration enhances the nitrifying reaction to remove the organic matter [32]. The trapping and removal of organic matter on top of the filter beds to be processed by earthworms and microbes could have also contributed to the increase in the DO. Earthworms increase the natural aeration by granulating the trapped oxygen depleting particles [9]. This could explain the further DO improvement in the vermifiltrates after the macrophyte chamber.

The high TDS, TSS, chlorides, turbidity and DOC recorded in the wastewater could be sourced from the presence of organic and inorganic solids [3]. These solids are either suspended or dissolved [34]. Turbidity represents the cloudiness of wastewater due to the presence of both macroscopic and microscopic suspended solids [35]. In the treated effluents, TDS, TSS, chlorides, turbidity and DOC were consistently reduced over the sampling period and this could be attributed to the actions of the macrophytes and earthworms. Macrophyte roots and vermibeds are capable of trapping the organic and inorganic solids where they are biologically processed by earthworms and microbes [16,17]. Macrophyte roots houses denitrifying bacteria and other microorganisms that assimilate organic matter using C as an electron donor [36]. The successful reduction of BOD and COD in the treated effluents has a direct link to the TDS, TSS, chlorides, turbidity and DOC removal [37]. The removal of BOD and COD rely largely on the good combination of the physical and microbial mechanisms [10]. The removal rates for BOD and COD in bio-filters are most likely associated with sedimentation of solids and their rapid decomposition [38]. Earthworms contribute to BOD and COD removal through enzymatic actions where the earthworms work as biological catalysts to biodegrade the solids [39,40]. This could explain the significant reduction in BOD and COD recorded in the MAV effluents when compared to the control. In most cases, there were no significant variations of BOD and COD removal between the MAV treatments and this demonstrated that the varied macrophyte and earthworm species were similar in their treatment performances.

Domestic sewage in septic tanks is one of the prevalent source of nitrogen and phosphorus contamination. If these nutrients are not removed in domestic sewage, discharge will cause excess algae growth in the discharged waterbodies [23]. The excess algae growth depletes oxygen and results in the death of fishes and other aquatic organisms. Macrophytes and earthworm species have several different properties that influence their treatment performances in nutrient removal. Variations in MAV treatment performances of the different macrophytes and earthworm species could be due to their specific physiological, morphological and biochemical structure [28]. Results revealed that the developed MAV systems efficiently removed nutrients from the wastewater and this have provided evidences that the developed MAV systems contributed considerably well to nutrient removal. [41] recorded similar results when macrophytes were used to treat wastewater in a constructed wetland. Processes that affect $N_{tot}$ removal in a MAV system includes volatilization, nitrification, denitrification, nitrogen fixation, plant and microbial uptake, mineralization (ammonification), fragmentation, sorption, desorption and leaching [42]. The possible influence of macrophytes in $N_{tot}$ removal mechanisms are direct uptake by root cells and denitrification process by Pseudomonas bacteria which are mostly attached to the roots [43]. $NO_3$-N concentrations were very low ($<1.0$ mgL$^{-1}$) in treated effluents, indicating that the first product of nitrification was quickly oxidized in the treatment process. A substantial proportion of the $NO_3$-N entering MAV is believed to undergo denitrification [28]. This view is based largely on the finding that $NO_3$-N disappears often due to denitrification. This supports the notion that denitrification rate is directly useful

for MAV design to optimize the removal of $NO_3$-N in the treatment process. [44] investigation of the nutrient distribution pathway and their removal efficiency concluded that macrophytes provide good growth conditions for microbes that aids in the nutrient removal. The vermifiltration removes nutrients through the action of earthworms [17]. The earthworms feed and assimilate the trapped nutrients for biomass gain and other physiological activities. The least removal of nutrients in the control could obviously be due to the absence of macrophytes and earthworms. Ammonia nitrogen includes both the ionized form ($NH_4$-N) and the unionized form ($NH_3$). An increase in pH favors the formation of the toxic $NH_3$, while a decrease favors the formation of the $NH_4$-N [45]. This could explain the direct increase in $NH_4$-N and a decrease in $NH_3$ when the pH reduced in the treated effluents. $NH_3$ is a common toxicant derived form in wastewater so removal in wastewater treatment is crucial. Aside the feacal sources, the presence of $P_{tot}$ in the wastewater might be attributed to the use of phosphate rich detergents that are washed into the septic tanks. [46] indicated that macrophytes, earthworms and microorganisms are important sinks for phosphorus, where they uptake and assimilate the nutrient for tissue development. This could explain the significant removal of $P_{tot}$ in the MAV effluents when compared to the control. The control recorded the least $P_{tot}$ removal and this could be due to the absence of a MAV system. Additionally, results showed no significant variations in all $P_{tot}$ removal performance between the MAV treatments and this apparently revealed that the varied macrophytes and earthworm species were similar in their treatment performances for $P_{tot}$ removal.

### 4.2. Pathogen removal

The coliform group, *E. coli* and helminths were used as indicators of pathogenic organisms due to their faecal origin that are of critical concern to public health [36]. These indicators were used to evaluate the pathogenic quality of the treated effluents. The characteristics of wastewater, with particular reference to pathogen indicators were high and this could be due to the influx of human feces into the septic tank [2]. Large populations of the pathogen indicators grow in the intestinal tracts of humans and are excreted in faecal wastes. However, in this study, these large pathogen counts were significantly reduced in the treated effluents. Sedimentation through filter beds, direct kill due to harsh environmental conditions created in waste treatment system, predation of the pathogen by earthworms, adsorption, physical straining, natural die-off and exposure to biocides excreted by macrophytes are possible factors that could have accounted for the reduction of pathogen load in the treated effluents [47]. This revealed that the least pathogen reduction that was recorded in the control might be due to the absence of the MAV system. Again, there was no significant differences in pathogen removal in the MAV treatments in most cases and this revealed that the different macrophytes and the earthworm species were similar in their treatment performances for pathogen removal. Attachment or adhesion of pathogens may take place on the surface of macrophytes as well as on the inner walls of the container of the wastewater. In the macrophyte chamber, sedimentation is expected to play a major role in pathogen removal because of the quiescent conditions that are created by the plant cover [1]. However, the pathogen removal appeared much rapid after vermifiltration compared to the macrophyte chamber effluent. This could be due to the additional pathogen removal by earthworms and microorganisms. Earthworms and microorganisms remove pathogens via biodegradation of organic matter and biological inactivation of pathogenic organisms' actions [44]. [29] reported that earthworms devour on the pathogens and promote the production of antibiotics that kill the pathogenic organisms. Thus, earthworms predate the pathogens which get filtered into the vermibed [16]. BOD and COD removal from the vermifiltration process also creates unsuitable physico-chemical

environment for pathogens to thrive and thus makes them more susceptible to die-off in the filter bed [48].

### 4.3. Vermiculture

Results of earthworm mortality recorded in the treatments are lower than that recorded in previous studies [9,16,17] and this could be due to the aerobic conditions caused by the presence of the macrophytes. Anaerobic and anoxic conditions are major sources of worm mortality Results from this study demonstrated that the vermibed in the MAV systems did not only facilitate the wastewater treatment process but also maintained the earthworm mortality below 10%. The activities of the earthworms in the vermibed improved oxygen penetration and created favorable conditions for the aerobic activity of microorganisms. This might be the reason for the elimination of the formation of odors and sludge [49]. Aside the suitable HLR used, this could also explain why there was no clogging in the vermibeds throughout the study. A vermifiltration system is ineffective when there is high earthworm mortality [15]. The growth and reproduction of earthworms increased at the end of the study and this is in accordance with previous studies [50–52]. Earthworms add to their body biomass through direct consumption, digestion and assimilation of the organic materials filtered on vermibeds [53]. However, the individual weight gain, biomass gain and individual growth rate were significantly higher in *Eudrilus eugeniae* treatments compared to *Eisenia foetida* and this could be due to the earthworm species-specific growth and feeding characteristics. The reproduction between the two earthworm species were not significantly different and results are again in accordance with previous studies [17,50–52].

### 5. Conclusion

Domestic wastewater from septic tanks in Ghana are poorly treated prior discharge. Installation of the developed MAV in this study could serve as a secondary treatment system to household septic tanks. This is because results from this study have revealed that the developed MAV systems were effective in the removal of solids and nutrients, (>60%) and pathogens (>90%) from domestic wastewater. Thus, the combined effect of macrophytes and the earthworms in the MAV systems provided ideal conditions that reduced solids, nutrients and pathogens from the wastewater. The existence of the macrophytes in the developed MAV did not only contributed to the treatment of the wastewater, but also improved the aeration of the system that promoted the vermiculture development for vermifiltration in the vermibed. Aside the biomass gain where *Eudrilus eugeniae* significantly recorded higher values than *Eisenia foetida*, the vermiculture of the two earthworm species were equally efficient in the vermifiltration process. In future research, more attention is needed to test the robustness of the developed MAVs in terms of harsh conditions such as high HLRs (>0.5 $m^3$ $m^{-2}$ $d^{-1}$), high intensity of contamination, cold climate, extreme pH and sodium toxicity.

### Supporting information

**S1 File. Study's underlying data set.**
(XLSX)

### Acknowledgments

Authors would like to acknowledge all technicians at the Environmental Quality Engineering (EQE) Laboratory of the Department of Civil Engineering, KNUST and Institute of Industrial Research who provided laboratory and technical assistance throughout the study.

## Author Contributions

**Conceptualization:** Rapheal Nsiah-Gyambibi.

**Data curation:** Rapheal Nsiah-Gyambibi.

**Formal analysis:** Rapheal Nsiah-Gyambibi.

**Funding acquisition:** Rapheal Nsiah-Gyambibi.

**Investigation:** Rapheal Nsiah-Gyambibi, Emmanuel Acheampong.

**Methodology:** Rapheal Nsiah-Gyambibi, Emmanuel Acheampong.

**Project administration:** Rapheal Nsiah-Gyambibi, Emmanuel Acheampong.

**Resources:** Rapheal Nsiah-Gyambibi, Emmanuel Acheampong, Elizabeth Von-Kiti.

**Software:** Rapheal Nsiah-Gyambibi, Emmanuel Acheampong, Elizabeth Von-Kiti, Christian Larbi Ayisi.

**Supervision:** Rapheal Nsiah-Gyambibi, Emmanuel Acheampong.

**Validation:** Rapheal Nsiah-Gyambibi, Emmanuel Acheampong, Elizabeth Von-Kiti, Christian Larbi Ayisi.

**Visualization:** Rapheal Nsiah-Gyambibi, Emmanuel Acheampong, Elizabeth Von-Kiti, Christian Larbi Ayisi.

**Writing – original draft:** Rapheal Nsiah-Gyambibi.

**Writing – review & editing:** Rapheal Nsiah-Gyambibi, Emmanuel Acheampong, Elizabeth Von-Kiti, Christian Larbi Ayisi.

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
