## [Decision Letter · Decision Letter 0]

21 Dec 2022

PONE-D-22-27352Performance evaluation of developed macrophyte-assisted vermifiltration system designed with varied macrophytes and earthworm species for domestic wastewater treatment.PLOS ONE

Dear Dr. Nsiah-Gyambibi,

Thank you for submitting your manuscript to PLOS ONE. After careful consideration, we feel that it has merit but does not fully meet PLOS ONE’s publication criteria as it currently stands. Therefore, we invite you to submit a revised version of the manuscript that addresses the points raised during the review process.

We look forward to receiving your revised manuscript.

Kind regards,

Frank O. Masese, Ph.D

Academic Editor

PLOS ONE

Journal Requirements:

"The project was funded by the Government of Ghana Research fund with the credit number 4646-GH. Authors would like to acknowledge all technicians at the Environmental Quality Engineering (EQE) Laboratory of the Department of Civil Engineering, KNUST who provided laboratory and technical assistance throughout the study"

"The project was funded by the Government of Ghana Research fund with the credit number 4646-GH"

3. "In your Data Availability statement, you have not specified where the minimal data set underlying the results described in your manuscript can be found. PLOS defines a study's minimal data set as the underlying data used to reach the conclusions drawn in the manuscript and any additional data required to replicate the reported study findings in their entirety. All PLOS journals require that the minimal data set be made fully available. For more information about our data policy, please see http://journals.plos.org/plosone/s/data-availability.

We will update your Data Availability statement to reflect the information you provide in your cover letter."

Reviewers' comments:

Reviewer's Responses to Questions

**Comments to the Author**

1. Is the manuscript technically sound, and do the data support the conclusions?

Reviewer #1: Yes

Reviewer #2: Yes

2. Has the statistical analysis been performed appropriately and rigorously? 

Reviewer #1: Yes

Reviewer #2: Yes

3. Have the authors made all data underlying the findings in their manuscript fully available?

Reviewer #1: Yes

Reviewer #2: Yes

4. Is the manuscript presented in an intelligible fashion and written in standard English?

Reviewer #1: Yes

Reviewer #2: Yes

5. Review Comments to the Author

Reviewer #1: Authors describe the use of various macrophytes and earthworm species in the effort for doing vermifiltration in WWTP. Several suggestions can be addressed below:

1. Abstract seems to be too qualitative. Authors are suggested to provide more quantitative insight.

2. Line 86 and below in introduction are suggested to be separated as new paragraph.

3. Experimental set up has no reference

4. Authors need to conduct various test such as homogenity, normality, and independency before performing anova

5. There are many inconsistencies in writings and also syntax errors. Authors are suggested to proofread the manuscript by the help of native English professional.

Reviewer #2: (1) On the Experimental setup

(a) It is not clear why a hydraulic loading rate (HLR) is provided or its significance

(b) It is not clear why the Effluent samples were collected every 48 hours and for two weeks

(c) It is not clear what the stage of growth / growth characteristics of the macrophytes were after establishment in the experimental treatment units

(2) On the results section on the expression of pathogen and helminth characteristics of wastewater and Pathogen count in effluents:

(a) The units are indicated as cfu100 mgL-1 . Why not the conventional CFU 100 ml-1.?

(b) Consider using the log unit for pathogen count – this is convenient when working with wastewater

6. PLOS authors have the option to publish the peer review history of their article (what does this mean?). If published, this will include your full peer review and any attached files.

Reviewer #1: No

Reviewer #2: **Yes: **Njenga Mburu

---

## [Author Response · Author response to Decision Letter 0]

4 Jan 2023

Rebuttal letter

Reviewers' comments:

Reviewer #1: Authors describe the use of various macrophytes and earthworm species in the effort for doing vermifiltration in WWTP. Several suggestions can be addressed below:

1. Abstract seems to be too qualitative. Authors are suggested to provide more quantitative insight.

Authors’ Response: Authors have provided Quantitative insights in the Abstract of the revised manuscript in Line 38, Line 39, Line 41 and Line 45. 

2. Line 86 and below in introduction are suggested to be separated as new paragraph.

Authors’ Response: Authors agree; Line 86 and below in introduction have been separated as new paragraph in the revised manuscript.

3. Experimental set up has no reference

Authors’ Response: References have been provided in the revised manuscript of the Experimental set up in Line 126, Line 128, Line 131 and Line 137. The references provided are; 

 1. Francesconi, K. A., Gailer, J., Edmonds, J. S., Goessler, W., & Irgolic, K. J. Uptake of arsenic-betaines by the mussel Mytilus edulis. Comparative Biochemistry and Physiology, 1999, 122, 131–137.

2. Yu, J., Xiong, M., Ye, S., Li, W., Xiong, F., Liu, J. and Zhang, T., Effects of stocking density and artificial macrophyte shelter on survival, growth and molting of jevenile red swamp crayfish (Procambarus clarkii) under experimental conditions. Aquaculture, 2020, 521, 735001.

4. Authors need to conduct various test such as homogenity, normality, and independency before performing anova.

Authors’ Response: Authors conducted chi-square tests of homogeneity and Shapiro-Wilk normality to determine the homogeneous, parametric and interdependency nature of the data, before performing ANOVA. This has been stated in the Statistical analysis section of the revised manuscript in Line 199 to Line 201. 

5. There are many inconsistencies in writings and also syntax errors. Authors are suggested to proofread the manuscript by the help of native English professional.

Authors’ Response: Authors have proofread the manuscript with the help of native English professional and have corrected all inconsistencies in writings and syntax errors. 

Reviewer #2: (1) On the Experimental setup

(a) It is not clear why a hydraulic loading rate (HLR) is provided or its significance

Authors’ Response: Authors have provided the significance of providing a hydraulic loading rate (HLR) in Line 144 to 146, and Line 444 to 445. 

(b) It is not clear why the Effluent samples were collected every 48 hours and for two weeks

Authors’ Response: Effluent samples were collected every 48 hours and for two weeks to provide the maximum stabilization conditions for effective analysis. This sampling procedure was used and established in Nsiah-Gyambibi et al. (2022) and Awuah (2006). These established references have been provided in Line 148 to 149 in the revised manuscript.

(c) It is not clear what the stage of growth / growth characteristics of the macrophytes were after establishment in the experimental treatment units

Authors’ Response: Authors have provided the stocking area of the macrophytes in Line 129 to 130. Authors however deemed the growth characteristics of the macrophytes had little contribution to the outcomes and purpose of this laboratory scaled study due to the sampling method chosen. Nevertheless, in our future and subsequent development of this study where direct field experiments will be combined with the laboratory scaled units in an extended sampling method, the growth characteristics of the macrophytes would be adequately investigated. 

(2) On the results section on the expression of pathogen and helminth characteristics of wastewater and Pathogen count in effluents:

(a) The units are indicated as cfu100 mgL-1 . Why not the conventional CFU 100 ml-1.?

Authors’ Response: Authors agree with reviewer. Authors have corrected all cfu100 mgL-1 into the conventional CFU 100 ml-1 in the revised manuscript. 

(b) Consider using the log unit for pathogen count – this is convenient when working with wastewater

Authors’ Response: Authors have considered using the log unit for pathogen count. All pathogen count have been changed into log unit in the revised manuscript.

---

## [Decision Letter · Decision Letter 1]

5 Feb 2023

Performance evaluation of developed macrophyte-assisted vermifiltration system designed with varied macrophytes and earthworm species for domestic wastewater treatment.

PONE-D-22-27352R1

Dear Dr. Nsiah-Gyambibi,

We’re pleased to inform you that your manuscript has been judged scientifically suitable for publication and will be formally accepted for publication once it meets all outstanding technical requirements.

Kind regards,

Frank O. Masese, Ph.D

Academic Editor

PLOS ONE

Additional Editor Comments (optional):

Reviewers' comments:

Reviewer's Responses to Questions

**Comments to the Author**

1. If the authors have adequately addressed your comments raised in a previous round of review and you feel that this manuscript is now acceptable for publication, you may indicate that here to bypass the “Comments to the Author” section, enter your conflict of interest statement in the “Confidential to Editor” section, and submit your "Accept" recommendation.

Reviewer #1: All comments have been addressed

Reviewer #2: All comments have been addressed

2. Is the manuscript technically sound, and do the data support the conclusions?

Reviewer #1: Yes

Reviewer #2: Yes

3. Has the statistical analysis been performed appropriately and rigorously? 

Reviewer #1: Yes

Reviewer #2: Yes

4. Have the authors made all data underlying the findings in their manuscript fully available?

Reviewer #1: Yes

Reviewer #2: Yes

5. Is the manuscript presented in an intelligible fashion and written in standard English?

Reviewer #1: Yes

Reviewer #2: Yes

6. Review Comments to the Author

Reviewer #1: (No Response)

Reviewer #2: (i) Authors have addressed and clarified issues raised during the review

(b) The work addresses the important issue of sustainable sanitation and pollution prevention in the developing countries

7. PLOS authors have the option to publish the peer review history of their article (what does this mean?). If published, this will include your full peer review and any attached files.

Reviewer #1: **Yes: **Setyo Budi Kurniawan

Reviewer #2: **Yes: **Dr. Njenga Mburu

---

## [Editor Report · Acceptance letter]

10 Feb 2023

PONE-D-22-27352R1 

Performance evaluation of developed macrophyte-assisted vermifiltration system designed with varied macrophytes and earthworm species for domestic wastewater treatment. 

Dear Dr. Nsiah-Gyambibi:

I'm pleased to inform you that your manuscript has been deemed suitable for publication in PLOS ONE. Congratulations! Your manuscript is now with our production department. 

Kind regards, 

on behalf of

Dr. Frank O. Masese 

Academic Editor

PLOS ONE